# Implementation of RT-RAA and CRISPR/Cas13a for an NiV Point-of-Care Test: A Promising Tool for Disease Control

**DOI:** 10.3390/v17040483

**Published:** 2025-03-27

**Authors:** Jingqi Yin, Jin Cui, Hui Zheng, Tingting Guo, Rong Wei, Zhou Sha, Shaopeng Gu, Bo Ni

**Affiliations:** 1College of Veterinary Medicine, Shanxi Agricultural University, Jinzhong 030031, China; yjqhghg@163.com; 2China Animal Health and Epidemiology Center, Qingdao 266011, China; cuijin@cahec.cn (J.C.); zhenghui@cahec.cn (H.Z.); weirong@cahec.cn (R.W.); 3College of Veterinary Medicine, Yangzhou University, Yangzhou 225001, China; ttguo1989@yzu.edu.cn

**Keywords:** Nipah virus, RT-RAA, CRISPR/Cas13a, lateral flow strips, point-of-care test

## Abstract

Nipah virus (NiV) is a severe zoonotic pathogen that substantially threatens public health. Pigs are the natural hosts of NiV and can potentially transmit this disease to humans. Establishing a rapid, sensitive, and accurate point-of-care detection method is critical in the timely identification of infected pig herds. In this study, we developed an NiV detection method based on reverse transcription–recombinase polymerase amplification (RT-RAA) and the clustered regularly interspaced short palindromic repeats (CRISPR)-associated protein 13a (Cas13a) system for the precise detection of NiV. The highly conserved region of the NiV gene was selected as the detection target. We first designed eleven pairs of RT-RAA primers, and the optimal primer combination and reaction temperature were identified on the basis of RT-RAA efficiency. Additionally, the most efficient crRNA sequence was selected on the basis of the fluorescence signal intensity. The results revealed that the optimal reaction temperature for the developed method was 37 °C. The detection limit was as low as 1.565 copies/μL. Specificity testing revealed no cross-reactivity with nucleic acids from six common swine viruses, including Seneca virus A (SVA), foot-and-mouth disease virus (FMDV), classical swine fever virus (CSFV), porcine epidemic diarrhea virus (PEDV), African swine fever virus (ASFV), and pseudorabies virus (PRV). A validation test using simulated clinical samples revealed a 100% concordance rate. The detection results can be visualized via a fluorescence reader or lateral flow strips (LFSs). Compared with conventional detection methods, this RT-RAA-CRISPR/Cas13a-based method is rapid and simple and does not require scientific instruments. Moreover, the reagents can be freeze-dried for storage, eliminating the need for cold-chain transportation. This detection technology provides a convenient and efficient new tool for the point-of-care diagnosis of NiV and for preventing and controlling outbreaks.

## 1. Introduction

Nipah virus (NiV) is an enveloped, negative-sense, single-stranded RNA virus belonging to the Paramyxoviridae family, the Paramyxovirinae subfamily, and the genus *Henipavirus*. It has a diameter of approximately 150 nm and a full genome length of approximately 18.2 kb that encodes six major structural proteins: fusion protein (F), glycoprotein (G), nucleocapsid protein (N), phosphoprotein (P), matrix protein (M), and RNA-dependent RNA polymerase (L), as well as three nonstructural proteins (V, W, and C proteins) [1]. NiV is an emergent zoonotic pathogen characterized by a high mortality rate and is classified at a biosafety level of 4. Owing to ineffective treatments and vaccine preparations, NiV has garnered significant international public health attention [2].

As natural hosts of NiV, flying foxes can carry the virus for a long time and transmit the virus to pigs and other domestic animals or people through their urine and feces [3]. The first NiV outbreak in Malaysia occurred in 1998, and 1.1 million pigs were culled, causing economic losses of 350 million to 400 million USD [4]. The epidemiological characteristics of Nipah virus are characterized by localized outbreaks and high pathogenicity. NiV has appeared in countries and regions such as Malaysia, Singapore, Bangladesh, and India. The most recent NiV epidemic broke out in Siliguri Town, Kerala, India (66 cases, fatality rate 68%), in September 2023 [5]. India and China are geographically adjacent and share long land borders, and the ecological environment in parts of southern China could provide the ideal living conditions for NiV-hosting flying foxes, increasing the potential risk of cross-border virus spread. Due to the close ecological contact between flying foxes and densely farmed pig populations, this interaction makes pigs a high-risk host for NiV infection. As an amplifying host, pigs can transmit NiV to humans, resulting in numerous deaths [6,7]. Therefore, developing convenient, sensitive, and accurate point-of-care test (POCT) methods for pig testing is critical in the timely identification of infected pig herds.

With the development of diagnostic technology, a variety of NiV detection methods have been established, including virus isolation, immunohistochemical detection, serology, RT–PCR, RT–qPCR, and loop-mediated isothermal amplification (LAMP) [8,9,10,11,12,13]. However, these methods have several common drawbacks, such as the need for good laboratory conditions, expensive instruments, and experienced technicians, and are difficult to apply to POCTs. The clustered regularly interspaced short palindromic repeats (CRISPR)-associated protein (Cas) system is an acquired immune system used by prokaryotes to resist invading foreign genetic elements in phages or plasmids. This defense mechanism is present in most bacteria and all archaea and has powerful gene-editing capabilities [14]. The Cas13a protein of the CRISPR system combines with its paired CRISPR RNA (crRNA) and target RNA to form a complex and undergo conformational changes, activating RNA enzymatic activity, which cleaves the single-stranded RNA probe and generates a fluorescent signal [15]. However, when the target RNA concentration is low, the collateral cleavage activity of the Cas protein significantly decreases, so increasing the amount of target nucleic acid is a key prerequisite for improving the application of this method [16]. In 2018, Zhang Feng et al. first demonstrated the joint use of recombinase-aided amplification (RAA) technology and CRISPR technology, effectively resulting in high amplification rates due to RAA and high specificity due to CRISPR [17,18]. Most importantly, this method can be applied to lateral flow strips (LFSs) so that it can be easily applied for POCTs.

In this study, an RT-RAA-CRISPR/Cas13a detection method for NiV was established with high sensitivity, specificity, and repeatability and was further applied to LFSs to provide a new POCT for the rapid diagnosis and control of NiV (Figure 1).

## 2. Materials and Methods

### 2.1. Design of RT-RAA Primers, CRISPR crRNAs, Droplet Digital PCR (ddPCR) Primers and Probes, and RNA Reporters

The complete gene sequences of 62 Nipah virus isolates (Appendix A) were obtained from the GenBank database (NCBI, http://www.ncbi.nlm.nih.gov/, accessed on 10 May 2024) and aligned using MEGA (version 11.0.13). The conserved gene was selected as the detection target region for the design of RT-RAA primers, crRNAs, and droplet digital PCR (ddPCR) primers and probes. Eleven pairs of conserved region gene primers were designed according to RT-RAA design principles, and the T7 promoter sequence (GAAATTAATACGACTCACTATAGGG) was added to the 5′ end of the RT-RAA forward primer to facilitate in vitro transcription. Five crRNAs were designed on the basis of the target fragment sequence amplified by RT-RAA and the characteristics of crRNA recognition by LwaCas13a. The ddPCR primers and probes were designed according to the target region. To detect this region, we designed two types of reporters: an RNA reporter (5′-FAM-UUUUUUUUUUU-3′-BHQ1) was used for fluorescence analysis, and an LFS RNA reporter (5′-FAM-UUUUUUUUUUU-3′-biotin) was used for LFS analysis. The target sequences, RT-RAA primers, CRISPR crRNAs, ddPCR primers and probes, and RNA reporters were synthesized by Sangon Biotech (Shanghai, China). The details of the target sequence, RT-RAA primers, CRISPR crRNA, ddPCR primers and probes, and RNA reporters are presented in Table 1.

### 2.2. Preparation of crRNAs

To prepare crRNAs, the oligonucleotide T7-crRNA-F, which served as the synthetic crRNA transcription template, were combined with crRNA-R (1-5) in their respective pairs. The resulting oligonucleotide chain combinations were then reconstituted in annealing buffer to achieve a concentration of 10 µM. Equal volumes of the mixtures were prepared, denatured at 95 °C for 5 min, and then allowed to slowly cool to 25 °C. The cooling process lasted no less than 45 min to facilitate the annealing and formation of double-stranded DNA. This double-stranded DNA was used as a template for in vitro transcription, and crRNA was synthesized in vitro at 37 °C for 2 h using a T7 High Yield RNA Transcription Kit (Vazyme, Nanjing, China). The transcription product was subsequently treated with DNase I at 37 °C for 1 h to degrade the dsDNA template. The crRNA was purified via RNA isolation beads (SyNTHGENE, Nanjing, China). The concentration of the purified crRNA product was measured with a NanoDrop spectrophotometer (Thermo Scientific, Waltham, MA, USA). The obtained crRNA was stored at −80 °C until being used.

### 2.3. Preparation of NiV Single-Strand RNA Template and Viral Genome of Other Swine Diseases

The highly conserved sequence obtained through screening was used as the target gene fragment for detection, and the positive plasmid pUC57-NiV was synthesized. The recombinant plasmid was constructed by Sangon Biotech (Shanghai, China). Using the linearized plasmid containing the target sequence, RNA was transcribed in vitro with a T7 High Yield RNA Transcription Kit (Vazyme, Nanjing, China) according to the manufacturer’s instructions. After transcription, the product was treated with DNase I and purified with RNA isolation beads (SyNTHGENE, Nanjing, China). The obtained NiV single-strand RNA (ssRNA)template was divided into aliquots and frozen at −80 °C until being used.

Nucleic acids from Seneca virus (SVA), foot-and-mouth disease virus (FMDV), classical swine fever virus (CSFV), porcine epidemic diarrhea virus (PEDV), African swine fever virus (ASFV), and porcine pseudorabies virus (PRV) were preserved by the China Animal Health and Epidemiology Center.

### 2.4. RT-RAA and Primer Screening

A total of eleven pairs of primers, including F1R1, F2R2, F3R3, F4R4, F5R5, F6R6, F7R7, F8R8, F9R9, F10R10A, and F10R10B (Table 1), were screened to find the optimal RT–RAA primer pair that produced the maximum amount of target amplicons. The RT-RAA reactions were conducted according to the manufacturer’s instructions. Each RT-RAA reaction was prepared as a 50 µL reaction and contained 25 µL reaction buffer, 4 µL forward and reverse primer combinations (at a concentration of 10 µM), and 11 µL nuclease-free water. The mixture was transferred to a microtube containing the pellet enzymes provided with the RT-RAA Kit (Jiangsu Quitian Gene Biotechnology Co., Ltd., Wuxi, China). To activate the reaction, 5 µL NiV ssRNA template and 5 µL magnesium acetate were dispensed into the tube cap and spun down to start the reaction. The tubes were incubated at 37 °C for 40 min, and the RT-RAA products obtained were subjected to 2% agarose gel electrophoresis to visualize the amplicon under an imaging system (Vilber Bio-Imaging Fusion FX6, Collegien, France). The DNA marker, 6× DNA loading buffer, and agarose were purchased from Sangon Biotech (Shanghai, China).

### 2.5. Determining the Optimal Temperature for RT-RAA

To determine the optimal reaction temperature, RT-RAA was performed at 37 °C, 38 °C, 39 °C, 40 °C, and 41 °C according to the recommended optimal temperatures for the RT-RAA reaction enzyme, and the optimal reaction temperature was selected by observing the results of 2% gel electrophoresis under an imaging system (Vilber Bio Imaging Fusion FX6, Collegien, France). The DNA marker, 6× DNA loading buffer, and agarose were purchased from Sangon Biotech (Shanghai, China).

### 2.6. Screening for crRNAs and CRISPR/Cas13a Detection

The optimized RT-RAA system was used to amplify the NiV ssRNA template, the five designed crRNAs were used for CRISPR/Cas13a detection, and ddH_2_O was used as a negative template control. After the reaction, the fluorescence intensity was observed, the fluorescence value was determined, and the optimal crRNA was selected.

We determined the optimal crRNA for the RT-RAA–CRISPR/Cas13a assay with a fluorescence RNA reporter. The CRISPR/Cas13a reaction system included 1 µL Cas13a reaction buffer, 1 µL crRNA (7.5 ng/µL), 2 µL Cas13a (GenScript), and 5 µL nuclease-free water and was allowed to react in a constant-temperature metal bath at 37 °C for 10 min. After the reaction, 5 μL Cas13a reaction buffer, 4 μL rNTP mix, 1 μL T7 transcriptase (50 U/μL), 2 μL RNase inhibitor, 5 μL RT-RAA product, 5 μL fluorescence reporter (10 µM), and nuclease-free water were added to each mixture, resulting in a final volume of 50 μL. The mixture was incubated at 37 °C for 30 min. A blue light transilluminator (Tiangen Biotech, Beijing, China) was used to observe the fluorescence signal, which was recorded with a smartphone camera. The fluorescence intensities were detected using 20 µL of the CRISPR/Cas13a system with a multifunction microplate reader (Shanpu, Shanghai, China).

### 2.7. Lateral Flow Assays

For lateral flow detection, the fluorescence reporter used in the CRISPR/Cas13a reaction system was replaced with a biotin reporter, and the other components were the same as in the previously mentioned reaction system. The CRISPR/Cas13a detection reaction products were diluted 1:1 in ddH_2_O, and then the commercial lateral flow strips (TOLOBIO, Shanghai, China) were dipped into the reaction solutions at room temperature for 10 min. Then, the strips were removed and imaged using a smartphone camera.

### 2.8. Calibration of the NiV ssRNA Template Copy Number Using ddPCR

The NiV ssRNA template was serially diluted 10-fold to create 12 dilutions ranging from 10^0^ to 10^−11^. A digital PCR instrument (Pilot Gene Technologies, Hangzhou, China) was used to calibrate the nucleic acid copy number for seven serial dilutions of the NiV ssRNA template, specifically from 10^−5^ to 10^−11^. The reaction protocol consisted of reverse transcription at 50 °C for 20 min, followed by predenaturation at 95 °C for 5 min, amplification at 95 °C for 15 s, and annealing at 60 °C for 30 s, over 45 cycles. The instructions of the multiplex one-step RT–qPCR probe kit (Yeasen Biotechnology, Shanghai, China) were followed. The reaction system was as follows: 10 µL 2× MP buffer, 1 µL enzyme mixture, 0.5 µL each upstream and downstream primers and fluorescence probes (10 µM) (see Table 1 for primer and probe sequences), 5 µL diluted NiV ssRNA template, and ddH_2_O were added to a total volume of 20 µL. The calibrated 10^0^–10^−11^ dilution of the NiV ssRNA template was divided into aliquots and frozen at −80 °C until being used.

### 2.9. Sensitivity Detection of RT-RAA-CRISPR/Cas13a

The calibrated 10^0^–10^−11^ dilution of the NiV ssRNA template was used as a template to perform the RT-RAA-CRISPR/Cas13a reaction to determine its sensitivity. The limit of detection (LOD) was determined by repeating the process three times.

### 2.10. Specificity Detection of RT-RAA-CRISPR/Cas13a

Common swine diseases including SVA, FMDV, CSFV, PEDV, ASFV, and PRV were used as templates, the NiV ssRNA template was used as the positive control, and ddH_2_O was used as the negative control. RT-RAA-CRISPR/Cas13a was tested separately to determine its specificity.

### 2.11. Detection of RT-RAA-CRISPR/Cas13a Using Simulated NiV Clinical Samples

To prepare simulated NiV clinical samples, a FineMag rapid magnetic bead viral DNA/RNA extraction kit (Jifan Biotechnology, Beijing, China) was used to extract RNA from 20 healthy pig sera stored at the China Animal Health and Epidemiology Center. The extracted RNA served as a simulated NiV-negative clinical sample. Then, an NiV ssRNA template with a concentration of approximately 100 copies/µL was mixed with random healthy individual pig serum RNA at a ratio of 1:1 to serve as a simulated NiV-positive clinical sample. Simultaneous reverse transcription–quantitative PCR (RT-qPCR) testing, using the same primers and probes as those used in ddPCR, as well as the same reaction system and protocol as those described in Section 2.8, was performed on these 20 simulated clinical samples as a control to compare with the RT-RAA-CRISPR/Cas13a results. A blind test of RT-RAA-CRISPR/Cas13a was conducted on simulated NiV clinical samples to assess the accuracy of RT-RAA-CRISPR/Cas13a for clinical sample detection.

### 2.12. Statistics and Reproducibility

Three independent technical replicates were conducted to generate fluorescence and colorimetric readouts. Statistical significance was analyzed by using Prism 10 (GraphPad Software, version 10.1.2). Differences were considered significant when *p* < 0.05 and are indicated as NS (not significant), * *p* < 0.05, ** *p* < 0.01, and *** *p* < 0.001.

## 3. Results

### 3.1. Screening of RT-RAA Primers

Based on the alignment results of 62 NiV whole-genome sequences (Appendix A), primers were designed based on a conserved gene sequence (Table 1). First, on the basis of the agarose electrophoresis results, F10R10A was selected from ten pairs of primers including F1R1, F2R2, F3R3, F4R4, F5R5, F6R6, F7R7, F8R8, F9R9, and F10R10A (Figure 2A). On the basis of the RT-RAA primer design principle, F10R10B was designed by shortening the product length. Using the calibrated 10^0^–10^−10^ diluted NiV ssRNA template as a template, RT-RAA was performed using F10R10A and F10R10B. ddH_2_O was used as a negative control to verify the optimization results (Figure 2B,C). The results showed that the optimized F10R10B primer pair had the best amplification efficiency and strong specificity. Therefore, the F10R10B primer set was selected for subsequent experiments.

### 3.2. Determining the Optimal RT-RAA Reaction Temperature

The NiV ssRNA template was used as a template, and the F10R10B primer set was used for RT-RAA detection. The results revealed that the amplified band was the brightest and the sole band detected when the reaction temperature was 37 °C. Therefore, 37 °C was selected as the optimal reaction temperature for RT-RAA (Figure 3).

### 3.3. Screening of crRNAs 

The five designed crRNAs were used for CRISPR/Cas13a detection, and ddH_2_O was used as a negative template control. The results revealed that the fluorescence intensity value of crRNA1 was significantly greater than that of the other groups, so crRNA1 was selected for subsequent experiments (Figure 4).

### 3.4. Sensitivity Detection for RT-RAA-CRISPR/Cas13a

To assess the sensitivity of the RT-RAA-CRISPR/Cas13a method, NiV ssRNA templates were subjected to tenfold serial dilution and calibrated using ddPCR. Following the dilution process, the copy numbers of the NiV ssRNA templates were determined to be as follows: 52,935.615 copies/μL (10^−5^), 1409.226 copies/μL (10^−6^), 94.775 copies/μL (10^−7^), 43.923 copies/μL (10^−8^), 18.746 copies/μL (10^−9^), 9.864 copies/μL (10^−10^), and 1.565 copies/μL (10^−11^). RT-RAA-CRISPR/Cas13a reactions were performed using calibrated 10^0^–10^−11^ dilutions of NiV ssRNA templates as templates.

The results revealed that the RT-RAA-CRISPR/Cas13a method could still detect fluorescence signals that were significantly different from those of the negative control group when the concentration of the NiV ssRNA template was as low as 1.565 copies/μL (Figure 5A,B). The sensitivity of the LFS assay was also consistent with that of the fluorescence assay, which was able to detect the NiV ssRNA template at a concentration of 1.565 copies/μL (Figure 5C). The experiment was repeated three times, and the results all showed the same pattern, indicating that the RT-RAA-CRISPR/Cas13a detection limit for the NiV was 1.565 copies/μL.

### 3.5. Specificity of RT-RAA-CRISPR/Cas13a

The RT-RAA-CRISPR/Cas13a detection method was evaluated for specificity using nucleic acids from common swine disease viruses as templates, including SVA, FMDV, CSFV, PEDV, ASFV, and PRV. Both fluorescence and lateral flow assays demonstrated that a positive signal was generated only when the template was the NiV ssRNA template, indicating that the RT-RAA-CRISPR/Cas13a method has high specificity (Figure 6).

### 3.6. Detection of the Application Effect of RT-RAA-CRISPR/Cas13a in Simulated NiV Clinical Samples

Ten simulated NiV-positive clinical samples and ten simulated NiV-negative clinical samples were subjected to blinded testing using RT-RAA-CRISPR/Cas13a. Both the fluorescence and lateral flow results, along with RT-qPCR results, indicated that ten out of the twenty reactions produced positive signals, which was consistent with the sample preparation (Figure 7). The RT-qPCR results are shown in Appendix A, and the statistics of simulated clinical sample detection are presented in Table 2. The coincidence rate in the results of RT-qPCR, NiV RT-RAA-CRISPR/Cas13a fluorescence, and NiV RT-RAA-CRISPR/Cas13a lateral flow assays for detecting simulated NiV clinical samples was 100% (20/20). The number of positive samples was identical, suggesting that RT-RAA-CRISPR/Cas13a has potential for use in clinical sample analysis.

## 4. Discussion

The World Health Organization listed NiV disease as one of ten potential high-risk infectious diseases that needed attention in 2018. With the accelerated development of globalization and trade, NiV may enter China through international transportation, the trade of animal products, or the movement of people, especially in areas adjacent to South Asia and Southeast Asia. As important hosts of NiV, pigs play a key role in the epidemiology of NiV [7]. However, the lives of humans and pigs overlap greatly, and pigs are major export commodities. Once NiV infection occurs, the virus may spread rapidly, triggering a devastating human pandemic. Globally, there is an urgent need to improve NiV infection responses and to ensure that readily accessible, real-time diagnostic reagents are available to quickly detect and control outbreaks.

As of now, a variety of detection techniques for NiV have been developed, such as the inspection and quarantine industry standard for NiV issued by the General Administration of Quality Supervision, Inspection and Quarantine of the People’s Republic of China in 2014, which identified RT–PCR and RT–qPCR methods as laboratory detection methods for NiV at border ports [19]. The minimum detectable limit for an RT–ddPCR assay developed by Jiangbing Shuai et al. is 6.91 copies/reaction [20]. Éric Bergeron et al. established a detection method based on a split NanoLuc biosensor for the rapid and sensitive detection of NiV antibodies; the sensitivity of the detection method was 98.6%, and the specificity was 100%; however, the efficacy of this method using early samples containing only IgM was limited [21]. These methods require expensive laboratory testing equipment and professional personnel to complete the operation. However, it is difficult for small and medium-sized pig farms in poor economic conditions to achieve the relevant testing conditions, and some large-scale farms also have problems such as multiple shared laboratories, untimely sample submission, and the untimely inspection of submitted samples, thereby delaying the timing of epidemic prevention and control.

SHERLOCK technology emerged in 2018. This method combines nucleic acid preamplification with CRISPR/Cas technology to improve the sensitivity and specificity of detection. This method can quickly and efficiently amplify target nucleic acids under constant-temperature conditions and uses the specific recognition of the target sequence by the Cas enzyme and the splicing of the reporter molecule to achieve sensitive detection [18]. The Cas13 protein is a unique effector protein in the CRISPR family. Unlike other common Cas proteins (such as Cas9 and Cas12), Cas13 mainly targets single-stranded RNA (ssRNA) rather than DNA. The Cas13 protein is divided into several subtypes, including Cas13a, Cas13b, Cas13c, and Cas13d, which are slightly different in sequence, structure, and function. Among these, the *Leptotrichia shahii* Cas13a protein is the most widely studied isoform with high collateral efficacy and high specificity [22]. Its mature research foundation and optimized experimental conditions make it simple to apply in the field of diagnosis, so this study used the Cas13a protein for subsequent research. To date, RT-RAA and CRISPR/Cas13a technologies have been used to establish detection methods for a variety of pathogens, such as avian influenza virus (AIV), dengue virus (DENV), and severe acute respiratory syndrome coronavirus 2 (SARS-CoV-2) [23,24,25].

This study combined RT-RAA with CRISPR/Cas13a technology to construct a new NiV detection method that can achieve the isothermal amplification of low-copy-number samples without the use of expensive equipment while ensuring high specificity and sensitivity and is easy to perform. Moreover, the reagents used in the detection method can be freeze-dried without relying on cold chain transportation, making the method suitable for practical clinical applications. The results of this method can be read in a variety of ways. A portable LFS can be used. If wet onsite weather affects the LFS, then a cheaper blue light lamp can also be used to observe the results, which provides a new method for the rapid onsite detection of NiV.

There are several limitations to this study. The described field diagnostic methods rely on an RNA extraction step, which may necessitate specific requirements for the configuration of laboratory equipment, thus limiting its wide application in resource-poor areas. Future research may focus on developing detection schemes that do not require RNA extraction to simplify the operation process and shorten the detection time. In addition, since BSL-4 experimental facilities may be limited in many epidemic outbreak areas, if the virus activity can be reduced by effective inactivation treatment after sample collection, then the detection process can be transferred to BSL-3 or BSL-2 laboratories, thereby improving the operability and practicability of the detection method. Future work is required to further verify its performance in analyzing complex biological samples, such as respiratory secretions, blood, urine, feces, and organ tissues, to expand its application scope and improve its reliability and sensitivity for actual diagnosis. In addition, given the significant similarities in clinical symptoms between NiV and Hendra virus (HeV), it is necessary to develop differential diagnostic methods for these two pathogens to achieve rapid and accurate differentiation and reduce treatment errors caused by misdiagnosis, in order to guide clinical intervention and public health response measures more effectively.

## 5. Conclusions

An RT-RAA-CRISPR/Cas13a rapid detection method for NiV was successfully established. This can be represented by a fluorescence method or a lateral flow strip. This method has high sensitivity, and the minimum detection limit is 1.565 copies/μL. The specificity is good, and there is no cross-reaction with SVA, FMDV, CSFV, PEDV, ASFV, or PRV. Detection is completed within 80 min. This method provides experimental data and technical means for clinical sample analysis, epidemiological investigations, and the development of disease monitoring technology for NiV.

## Figures and Tables

**Figure 1 viruses-17-00483-f001:**
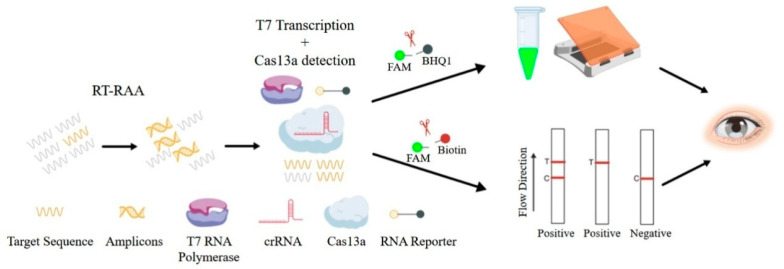
Schematic workflow of RT-RAA-CRISPR/Cas13a-based detection for NiV. The target sequence was amplified by RT-RAA, and the products were tested via the CRISPR/Cas13a detection system. After the Cas13a-crRNA complex is bound to the target RNA, the RNA reporter is cleaved. This cleavage generates a signal that can be observed directly by the naked eye as green fluorescence under blue light, or it can be measured using an LFS.

**Figure 2 viruses-17-00483-f002:**
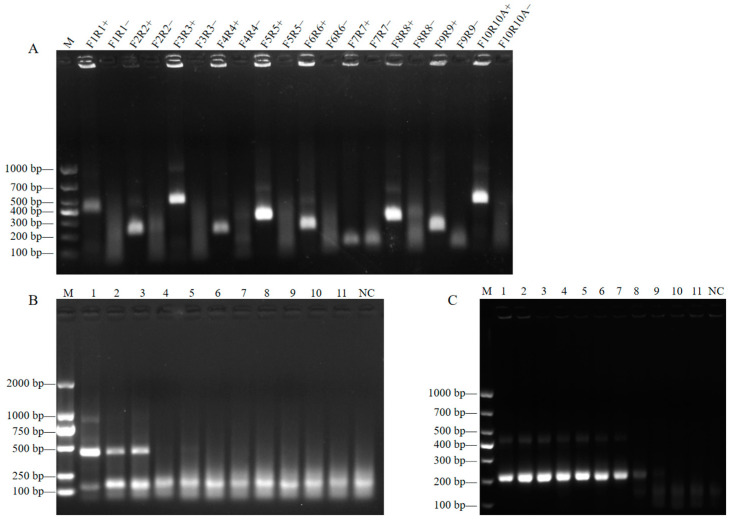
The 2% agarose gel electrophoresis analysis of the RT-RAA products. (**A**) Screening of the F1R1, F2R2, F3R3, F4R4, F5R5, F6R6, F7R7, F8R8, F9R9, and F10R10A primer pairs. The names of the primer pairs are indicated at the top. The sizes of the DNA ladder bands are shown on the left (bp); M, DNA marker DL1000. (**B**) The amplification efficiency of primer pair F10R10A was determined. 1–11: Template dilution ranges from 10^0^–10^−10^. (**C**) The amplification efficiency of primer F10R10B was determined. 1–11: Template dilution ranges from 10^0^–10^−10^. NC, negative control.

**Figure 3 viruses-17-00483-f003:**
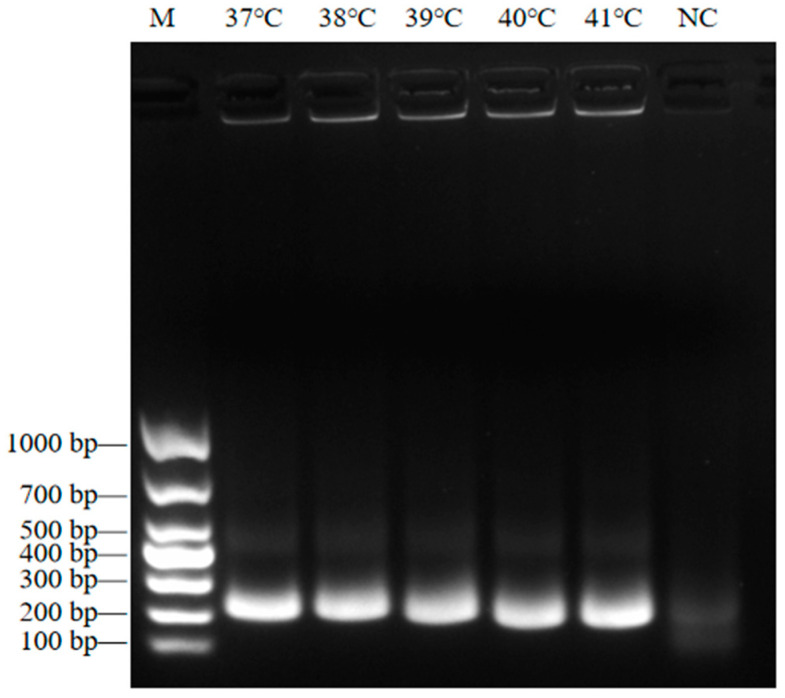
The 2% agarose gel electrophoresis analysis to determine the optimal RT-RAA reaction temperature. The reaction temperature is marked at the top. The size of the DNA ladder is shown on the left (bp); M, DNA marker DL1000. NC, negative control.

**Figure 4 viruses-17-00483-f004:**
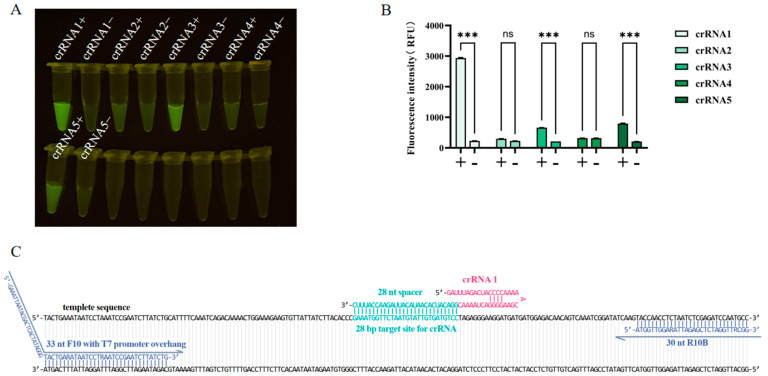
Optimization of crRNA for the RT-RAA–CRISPR/Cas13a assay. The CRISPR/Cas13a assay was performed using five crRNAs with the same template concentrations. + indicates NiV ssRNA template. − indicates the negative template control (nuclease-free water). (**A**) Fluorescence was imaged under a blue light transmission instrument using a smartphone camera. (**B**) The intensities of the fluorescence signals were determined with a microplate reader. Each column indicates the mean of triplicate fluorescence values ± SDs, and *** indicates a significant difference (*p* < 0.001); ns denotes no significant difference (*p* > 0.05). (**C**) Detailed sequence information for primers F10R10B and crRNA 1 targeting the template sequence.

**Figure 5 viruses-17-00483-f005:**
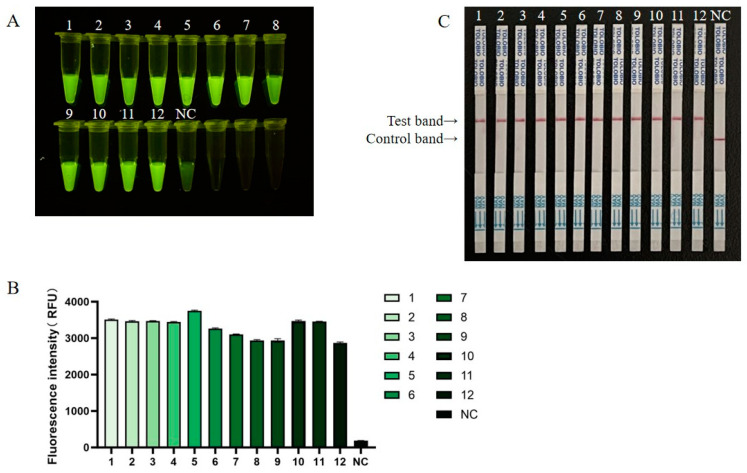
The sensitivity of RT-RAA-CRISPR/Cas13a. 1–12: Template dilution range from 10^0^–10^−11^. NC, negative control. (**A**) Fluorescence was imaged using a blue light transmission instrument using a smartphone camera. (**B**) The intensities of the fluorescence signals were determined using a microplate reader. Each column indicates the mean of triplicate fluorescence values ± SDs. (**C**) Sensitivity examination via the RT-RAA-CRISPR/Cas13a LFS assay.

**Figure 6 viruses-17-00483-f006:**
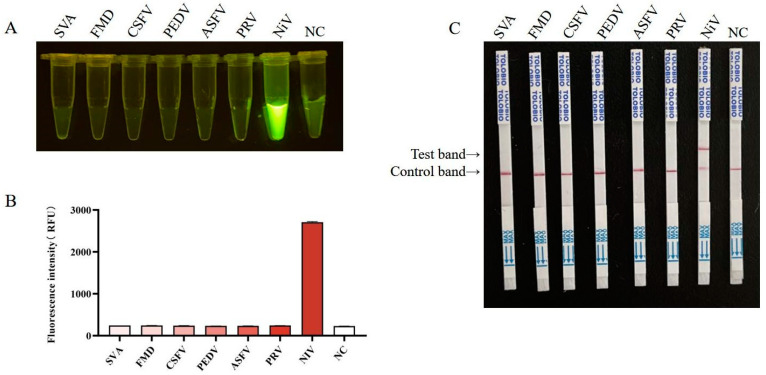
Specificity of RT-RAA-CRISPR/Cas13a. (**A**) Fluorescence was imaged using a blue light transmission instrument using a smartphone camera. (**B**) Intensities of the fluorescence signals were determined using a microplate reader. (**C**) Specificity examination via the RT-RAA-CRISPR/Cas13a LFS assay. NC, negative control.

**Figure 7 viruses-17-00483-f007:**
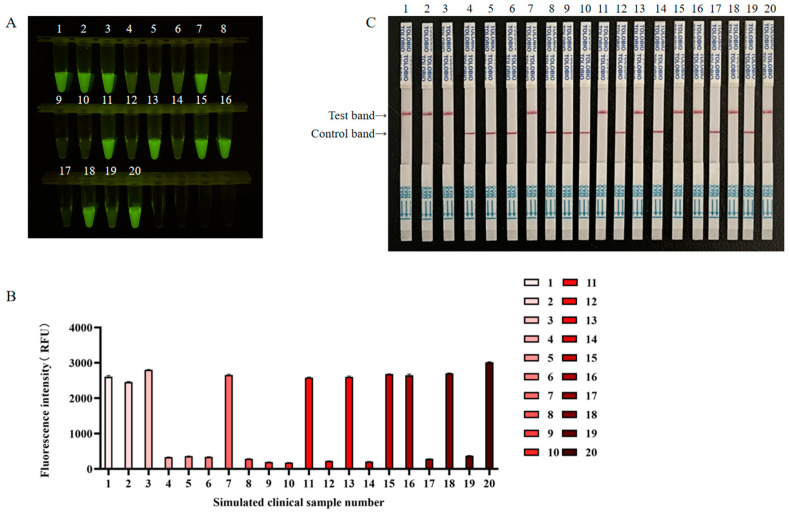
Simulated NiV clinical sample analysis by RT-RAA-CRISPR/Cas13a. (**A**) Fluorescence was imaged using a blue light transmission instrument using a smartphone camera. (**B**) Intensities of the fluorescence signals were determined using a microplate reader. (**C**) RT-RAA-CRISPR/Cas13a lateral flow assay detection results for simulated clinical samples.

**Table 1 viruses-17-00483-t001:** Target sequences, reverse transcription–recombinase polymerase amplification (RT-RAA) primers, CRISPR RNAs (crRNAs), droplet digital PCR (ddPCR) primers and probes, and RNA reporters.

Name	Sequence (5′—3′)
Target sequence	TACTGAAATAATCCTAAATCCGAATCTTATCTGCATTTTCAAATCAGACAAAACTGGAAAGAAGT GTTATTATCTTACACCCGAAATGGTTCTAATGTATTGTGATGTCCTAGAGGGAAGGATGATGATG GAGACAACAGTCAAATCGGATATCAAGTACCAACCTCTAATCTCGAGATCCAATGCC
NiV-F1	GAAATTAATACGACTCACTATAGGGAGATTATTTAACATGTACAGATCATATTTCGGA
NiV-R1	TTATGATGGTGTCTCTTAATGTATTGATATCTG
NiV-F2	GAAATTAATACGACTCACTATAGGGAATAATTAATATACATGAGTGTAGGCGATTAGG
NiV-R2	CAGATAAGATTCGGATTTAGGATTATTTCAGTA
NiV-F3	GAAATTAATACGACTCACTATAGGGCAATTTCTTTTTACAATTCAGGTATAGATGGAG
NiV-R3	TACTTCAGTAGAATCTGGATTACTATATACTGG
NiV-F4	GAAATTAATACGACTCACTATAGGGATGTTAAATGAGGCTATGAATTATTTTGATGAC
NiV-R4	TCTTCTGATGTTATTTTTGATGTGATAAAGTTC
NiV-F5	GAAATTAATACGACTCACTATAGGGCTAAAACAATTAAAAATATCACAGCAAGGACTA
NiV-R5	ATCATGATGAGATAATCTAGAAACTAACTTTGG
NiV-F6	GAAATTAATACGACTCACTATAGGGAGAATCATAAATATAGAAGGATAGGTCTCAACT
NiV-R6	ATCTCCATCTATACCTGAATTGTAAAAAGAAAT
NiV-F7	GAAATTAATACGACTCACTATAGGGTTAATCACAGAATTTCTAATAGTTGATCCTGAA
NiV-R7	GAATTTCAAAGGCCCATTTTATTGATATAGATT
NiV-F8	GAAATTAATACGACTCACTATAGGGAAGAGTCTATACATATTAAGACAATCCAAACAG
NiV-R8	ATCTCTTCTATCTTGAGTTATGAGATTTCTAGT
NiV-F9	GAAATTAATACGACTCACTATAGGGACTCTTTTAATAAGGTTAAATCTGCTCTCAATA
NiV-R9	GTATTGTCTGATTTATAGTGATTGTGAGGATTA
NiV-F10	TACTGARATAATCCTRAATCCGAATCTTATCTG
NiV-R10A	GAATGGATAAAAGATCATCAATAAACATAGACC
NiV-R10B	GGCRTTGGATCTCGAGATTARAGGTTGGTA
T7-crRNA-F	GAAATTAATACGACTCACTATAGGG
crRNA-R1	GATTTAGACTACCCCAAAAACGAAGGGGACTAAAACGGACATCACAATACATTAGAACCATTTC
crRNA-R2	TAGAGGTTGGTACTTGATATCCGATTTGGTTTTAGTCCCCTTCGTTTTTGGGGTAGTCTAAATCCCCTATAGTGAGTCGTATTAATTTC
crRNA-R3	TAGCCCCCAGAGGGCATTGGATCTCGAGGTTTTAGTCCCCTTCGTTTTTGGGGTAGTCTAAATCCCCTATAGTGAGTCGTATTAATTTC
crRNA-R4	TATCATAGACACTATATTGTAAATTCTGGTTTTAGTCCCCTTCGTTTTTGGGGTAGTCTAAATCCCCTATAGTGAGTCGTATTAATTTC
crRNA-R5	CAGGATCCTAGCCTCATCCTTGAGTTGGGTTTTAGTCCCCTTCGTTTTTGGGGTAGTCTAAATCCCCTATAGTGAGTCGTATTAATTTC
ddPCR-F	GATGATGGAGACAACAGTCAAATC
ddPCR-R	GACAGGGAACAAGGGATCAA
ddPCR-probe	ACCTCTAATCTCGAGATCCAATGCCCT
Fluorescence RNA Reporter	5′-FAM/UUUUUUUUUUU/BHQ1-3′
LFS RNA Reporter	5′-FAM/UUUUUUUUUUU/biotin-3′

**Table 2 viruses-17-00483-t002:** Simulated clinical sample detection result statistics.

Method	The Number of Positive Samples	The Number of Negative Samples	Total	Positivity Rate
RT-qPCR	10	10	20	50%
NiV RT-RAA-CRISPR/Cas13a fluorescence	10	10	20	50%
NiV RT-RAA-CRISPR/Cas13a lateral flow assays	10	10	20	50%
Coincidence rate	100%	100%	-	

## Data Availability

The data presented in this study are available on request from the corresponding author.

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
