# Peer review of "Implementation of RT-RAA and CRISPR/Cas13a for an NiV Point-of-Care Test: A Promising Tool for Disease Control"

_viruses, 2025, doi:10.3390/v17040483_

Round 1
Reviewer 1 Report
Comments and Suggestions for Authors
The manuscript described a nucleic acid detection method based on reverse transcription-recombinase polymerase amplification (RT-RAA) and the clustered regularly interspaced short palindromic repeats (CRISPR)/Cas13a for Nipah virus (NiV) point-of-care test. There are some major concerns to be considered before accept for publication.
Major concerns:
- Selection of the target gene of Niv is the key to the CRISPR/Cas13a detection method, espcially its variation among circulating strains. The authors need provide more data and information to explain how or why they chose this target sequcece of L gene for CRISPR/Cas13a, and discuss in the discussion section.
- As for the detection sensitivity, only using the standard plasmid DNA could not provide solid data to evaluate the sensitivity of RT-RAA-CRISPR/Cas13a. It is suggested that the authors provide more data of detecting the inactivated Niv or its genomes. In additon, the fluorescence intensity (A and B) and test line color (C) in Figure 5 have no signifidcant changes as concertration of the DNA template decreased by ten-fold. Could the authors explain the reasons? OR they are false detection signals?
- In detection of the simulated NiV clinical samples, it is necessary to compare with the standard reference method such as RT-qPCR. In additioin, the inactivated NiV culture or genomes are suggested using as the simulated samples.
Minor concerns:
- Line 89, Table 2 is not found in the manuscript.
- Lines 250-251 Figure 4. Please double check the “****”statistical annotation on crRNA2 in Figure 4B.
Reviewer 2 Report
Comments and Suggestions for Authors
This paper describes the development and evaluation of a Nipah Virus Point-of-Care test. This is interesting and important work to develop a much-needed method for clinical sample analysis & disease monitoring. The manuscript is well written and organised in a logical manner. I think your results are very promising.
Author Response
Comments 1: This paper describes the development and evaluation of a Nipah Virus Point-of-Care test. This is interesting and important work to develop a much-needed method for clinical sample analysis & disease monitoring. The manuscript is well written and organised in a logical manner. I think your results are very promising.
Response 1: Thank you for your nice and encouraging comment, we really appreciate it.
Reviewer 3 Report
Comments and Suggestions for Authors
This study combines RT-RAA and CRISPR/Cas13a technologies to develop a rapid and highly sensitive on-site detection method for Nipah virus (NiV), which has important potential for public health applications. The research design is reasonable, and the data generally support the conclusions. However, some experimental details need to be further supplemented, the result analysis needs to be more rigorous, and the discussion part needs to be more in-depth. The following are the specific opinions:
1 Table 1 does not show the probes labeled with ddPCR-probe.
2 Standardize the writing form of primers and choose either F1R1 or F1/R1.
3 Figure 2A is not the original figure and a part of it is missing. Please supplement it to make it complete.
4 The processes of primer screening and temperature optimization solely rely on the assessment of gel electrophoresis brightness, without the support of quantitative data. In future experimental designs, it is highly recommended to utilize fluorescent dyes for more accurate primer screening and temperature optimization, as this approach can provide more objective and reliable data for the evaluation of primer performance and reaction conditions.
Author Response
Comments 1: Table 1 does not show the probes labeled with ddPCR-probe.
Response 1: Thank you for your comment. I have included the ddPCR-probe sequence information in Table 1.
Comments 2: Standardize the writing form of primers and choose either F1R1 or F1/R1.
Response 2: Thank you for your comment. I have standardized the writing form of the primers to F1R1 as suggested. This change has been made consistently throughout the manuscript.
Comments 3: Figure 2A is not the original figure and a part of it is missing. Please supplement it to make it complete.
Response 3: Thank you for your comment. We have replaced Figure 2 and Figure 3 with the complete original nucleic acid gel images, as requested. The updated figures have been re-uploaded.
Comments 4: The processes of primer screening and temperature optimization solely rely on the assessment of gel electrophoresis brightness, without the support of quantitative data. In future experimental designs, it is highly recommended to utilize fluorescent dyes for more accurate primer screening and temperature optimization, as this approach can provide more objective and reliable data for the evaluation of primer performance and reaction conditions.
Response 4: Thank you for your suggestion. The current process of primer screening and temperature optimization relies solely on gel electrophoresis brightness, which lacks quantitative data. In future experimental designs, we will incorporate the use of fluorescent dyes for more accurate primer screening and temperature optimization. We really appreciate your valuable recommendation which will dramatically improvements our subsequent experiments.
Round 2
Reviewer 1 Report
Comments and Suggestions for Authors
No comments.